# Aptamer-assisted tumor localization of bacteria for enhanced biotherapy

Zhongmin Geng[1], Zhenping Cao[1], Rui Liu[1], Ke Liu[1], Jinyao Liu [1,2✉] & Weihong Tan [1]

Despite bacterial-mediated biotherapies have been widely explored for treating different types of cancer, their implementation has been restricted by low treatment efficacy, due largely to the absence of tumor-specific accumulation following administration. Here, the conjugation of aptamers to bacterial surface is described by a simple and cytocompatible amidation procedure, which can significantly promote the localization of bacteria in tumor site after systemic administration. The surface density of aptamers can be easily adjusted by varying feed ratio and the conjugation is able to increase the stability of anchored aptamers. Optimal bacteria conjugated with an average of $2.8 \times 10^5$ aptamers per cell present the highest specificity to tumor cells in vitro, separately generating near 2- and 4-times higher accumulation in tumor tissue at 12 and 60 hours compared to unmodified bacteria. In both 4T1 and H22 tumor-bearing mouse models, aptamer-conjugated attenuated Salmonella show enhanced antitumor efficacy, along with highly activated immune responses inside the tumor. This work demonstrates how bacterial behaviors can be tuned by surface conjugation and supports the potential of aptamer-conjugated bacteria for both targeted intratumoral localization and enhanced tumor biotherapy.

[1] Shanghai Key Laboratory for Nucleic Acid Chemistry and Nanomedicine, Institute of Molecular Medicine, State Key Laboratory of Oncogenes and Related Genes, Renji Hospital, School of Medicine, Shanghai Jiao Tong University, 200127 Shanghai, China. [2] Shanghai Key Laboratory of Gynecologic Oncology, Renji Hospital, School of Medicine, Shanghai Jiao Tong University, 200127 Shanghai, China. ✉email: jyliu@sjtu.edu.cn

In the late nineteenth century, W. Busch and W. Coley have independently utilized *Streptococcus pyrogens* and *Serratia marcescens* to treat cancer, which appears as distinct tumor suppression effects in some sarcoma patients[1,2]. Since then, the exploration of bacterial-based tumor therapies has never stopped, although the therapeutic outcomes of these treatments remain unstable[3]. Particularly, with recent progress in the fields of immunology and biotechnology, bacteria have attracted increasing attention in tumor biotherapy, both as immune modulators and drug delivery vehicles[4,5]. The infection by bacteria is able to promote the immunogenicity of tumors, which further induces effective antitumor immune responses[6]. As reported, different immune cell subtypes, such as CD4+ and CD8+ T cells, regulatory T cells, and tumor-associated macrophages, can be modulated to generate antitumor immunity by attenuated bacteria[7]. Furthermore, the colonization of bacteria in tumor tissue is capable to trigger the activation of innate immunocytes and the release of proinflammatory factors, which cause the disruption of tumor vasculature as well as the associated thrombosis[8]. Regarding the application of bacteria as drug carriers, the greatest strength is their favorable colonization in the tumor sites, which is attributed to the intratumor anaerobic condition, eutrophication, and the immunosuppressive microenvironment[9–11]. Namely, the use of bacteria as delivery vehicles can improve the accumulation and penetration of drugs in tumor tissue and hence maximize their therapeutic efficacies. Unfortunately, bacterial-mediated tumor therapy always suffers from low inhibition efficiency of tumor growth and undesirable dose-dependent side effects, as confirmed in the failure of phase 1 clinical trial of attenuated *Salmonella typhimurium*[12].

Substantial efforts have been made to tackle these challenges by modifying bacteria either chemically or genetically[13,14]. To reduce systemic toxicity, several bacterial species including *Listeria*, *Escherichia coli*, and *Clostridium* have been attenuated by removing key virulence factor genes[15]. Alternatively, decoration with various types of substrates on the bacterial surfaces, such as synthetic nanoparticles and therapeutic drugs, has been exploited to enhance antitumor activity[16–19]. We have recently wrapped with extra coatings to retain bacterial viability under unfriendly external conditions[20–24]. For instance, bacteria decorated with a stealth coating can decrease their elimination by macrophages, which subsequently prolong blood circulation following systemic injection[25]. However, previous modifications by both chemical and genetic approaches lack the ability to endow bacteria with increment in tumor-specific localization after administration. Therefore, strategies are highly desirable for increasing the tumor-targeting capacity of bacteria, which can in turn increase the safety and antitumor efficacy of bacterial-based therapy.

Aptamers, which are considered "chemical antibodies", have been broadly applied as targeting ligands due to their ability to specifically recognize cell-surface targets (such as nucleic acids and proteins), whole cells, and even tissues[26–28]. Similar to antibodies, aptamers exhibit many favorable features including high affinity, excellent specificity, and low immunogenicity[29]. In addition, aptamers can be easily synthesized in vitro through mature solid-state synthetic technology and be chemically modified with various functional groups[30]. By virtue of these unique characteristics, several therapeutic aptamers have been evaluated in clinical trials for their potential in cancer treatment[31]. However, aptamer-based drug delivery has been mainly focused on developing aptamer-drug conjugates and aptamer-functionalized nanoparticles, with aims to enhance antitumor efficacies by increasing the accumulation of drugs in tumors[32–34]. To the best of our knowledge, the use of aptamers to actively deliver living cells to the tumor site by a specific ligand−receptor interaction has not been reported yet.

In this work, we describe aptamer-assisted tumor localization of bacteria for enhanced biotherapy (Fig. 1a, b). Tumor-specific aptamers have been conjugated to the surface of bacteria by a single-step amidation procedure, which produces aptamer-conjugated bacteria (ApCB). The preparation shows a negligible influence on bacterial viability, as the obtained ApCB grows similarly to unmodified bacteria. The grafting density of aptamers on the bacteria surface can be readily tuned from $0.7 \times 10^5$ to $5.7 \times 10^5$ aptamers per cell by controlling the feed ratio. More importantly, the attachment is capable of decreasing the enzymatic hydrolysis of aptamers anchored on the surface, up to 80% of which remains intact after 24 h exposure to serum. Among a set of ApCB with different grafting densities, bacteria decorated with an average of $2.8 \times 10^5$ aptamers per cell exhibit the highest specificity to tumor cells in vitro, which separately achieves 2- and 4-fold higher colonization in tumor tissue at 12 and 60 h post tail vein injection in mice. As a proof-of-concept study, aptamer-conjugated attenuated *Salmonella Typhimurium VNP20009* (VNP) display significantly enhanced antitumor efficacy in both 4T1 and H22 tumor-bearing mouse models, accompanied with remarkably elicited immune responses within the tumor. Therefore, our work illustrates a strategy to manipulate bacterial behaviors by simple surface modification and demonstrates the promising of ApCB for enhanced bacterial-mediated cancer therapy.

## Results

**Preparation and characterization of ApCB.** By amide condensation, aminated AS1411 was linked to the carboxyl group of *N*-acetylmuramic acid (Fig. 1a), which extensively exists in the cell wall of Gram-negative bacteria and shows higher reactivity than other carboxyl groups from glutamate or aspartate residues[35–37]. With the help of 1-ethyl-3-(3-dimethylaminopropyl)carbodiimide (EDC) and *N*-hydroxysuccinimide (NHS)[38], ApCB were prepared by co-incubating *Escherichia coli Nissle* 1917 (EcN) with AS1411 in phosphate-buffered saline (PBS) at room temperature for 3 h and the resulting bacteria were purified simply by centrifugation. To enable the measurement by laser scanning confocal microscopy (LSCM) and flow cytometry, AS1411 labeled with fluorochrome Cyanine5 (Cy5) at 5′ end of the sequence was used to prepare ApCB. As visualized in Fig. 1c, LSCM images showed that EcN expressing green fluorescent protein (GFP) was efficiently decorated with AS1411 (red) under the experimental condition. Successful decoration with AS1411 was further confirmed by flow cytometric analysis. The mean fluorescence intensity of ApCB was around ten times higher than that of EcN (Supplementary Fig. 1).

A given amount of $1 \times 10^8$ colony-forming units (CFU) of EcN were incubated with 0, 2, 5, and 10 nmol of Cy5-labeled AS1411, which were termed as $EcN_{Cy5}$, $2ApCB_{Cy5}$, $5ApCB_{Cy5}$, and $10ApCB_{Cy5}$, respectively. Then, the resulting bacteria were analyzed by flow cytometry. As expected, the fluorescence signals increased with the concentration of AS1411, as revealed clearly by a continuous shift of the curve to higher intensity (Fig. 1d, e). To calculate the number of grafted aptamers per cell, EcN conjugated with different concentrations of Cy5-labeled AS1411 was detected by fluorescence spectrophotometer. As illustrated in Fig. 1f, the quantities of AS1411 conjugated on the surface of $2ApCB_{Cy5}$, $5ApCB_{Cy5}$, and $10ApCB_{Cy5}$ were calculated to be $0.7 \times 10^5$, $2.8 \times 10^5$, and $5.7 \times 10^5$ aptamers per cell, respectively.

To investigate whether the preparation procedure and the conjugated aptamers had toxic side effects on bacterial viability, the growth of ApCB was assessed using plate counting. Serial dilutions of EcN, 2ApCB, 5ApCB, and 10ApCB were prepared and then spread onto Luria Bertani (LB) agar plates. After incubation at 37 °C for 24 h, the viability of ApCB was determined by recording the

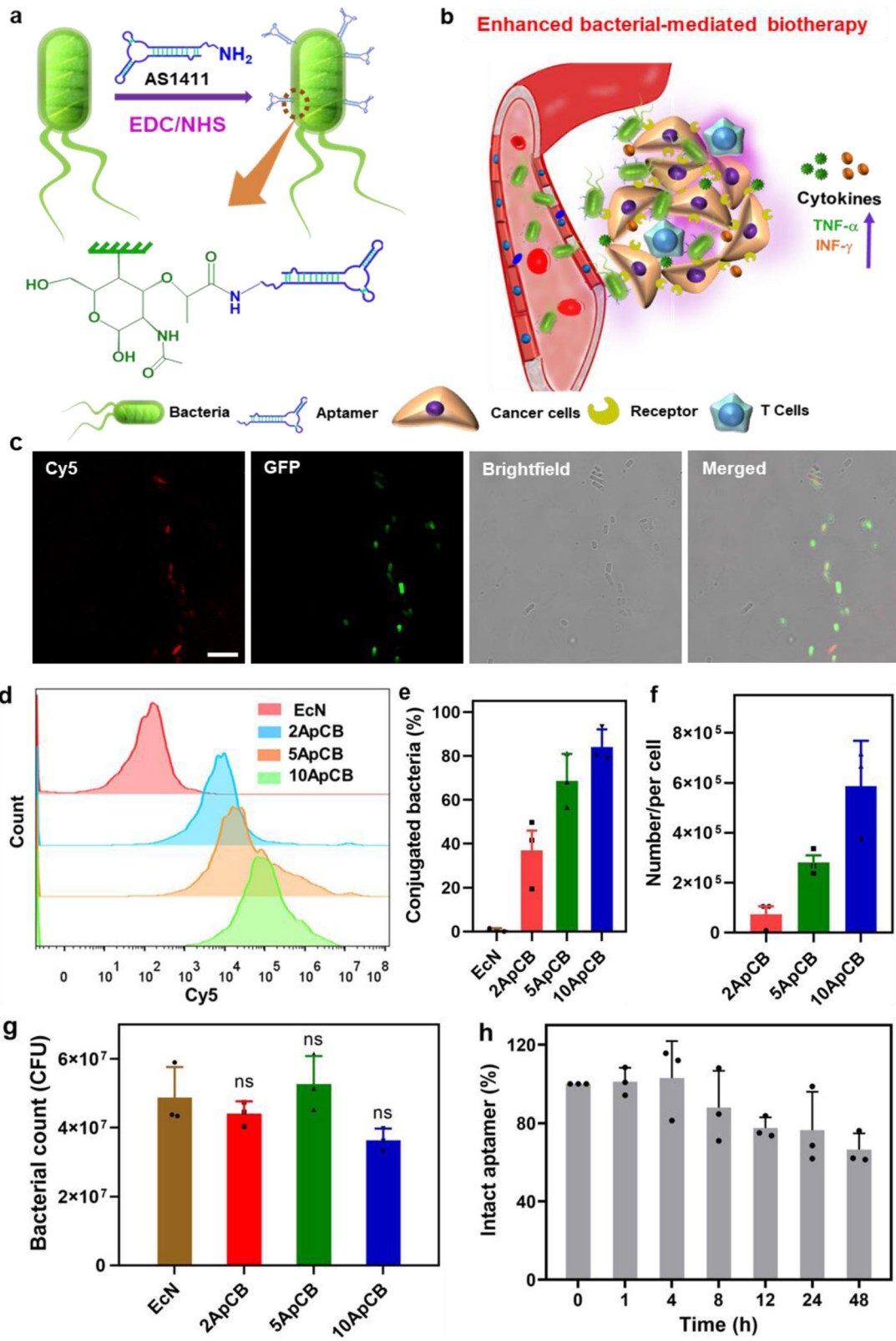

number of bacterial colonies. As shown in Fig. 1g, in contrast to EcN, 2ApCB, 5ApCB, and 10ApCB displayed no notable difference in colony number, verifying the uninfluenced bacterial viability after conjugation. In addition, the serum tolerance of an aptamer-based therapeutic agent is critical for in vivo applications[39]. We speculated that the conjugation of aptamers on the surface of bacteria could largely increase the steric hindrance towards the enzymatic

hydrolysis of AS1411 by nucleases[40]. To demonstrate this hypothesis, the stability of aptamers decorated on EcN was monitored by tracking the fluorescence intensity of Cy5-labeled AS1411. ApCB$_{Cy5}$ was analyzed by flow cytometry after incubation with 10% fetal calf serum in PBS at 37 °C for the predetermined time points. On the basis of our previous report, more than 60% of free AS1411 was found to be rapidly degraded after 48 h

**Fig. 1 Design, preparation and characterization of ApCB. a** Preparation of ApCB through amide condensation. **b** Aptamer-assisted tumor localization of bacteria for enhanced biotherapy. **c** Typical LSCM images of aptamer-conjugated bacteria. The red and green channels indicate aptamer conjugated with Cy5 and EcN producing GFP, respectively. Images are representative of three independent biological samples. Scale bar: 10 μm. **d** Flow cytometric analysis of EcN and EcN conjugated with Cy5-labeled AS1411. **e** Percentages of conjugated EcN under different feed ratios. Error bars represent the standard deviation ($n = 3$ independent experiments). Data are presented as mean values ± SD. **f** Average binding number of aptamers on each bacterial quantified by calculating the difference of fluorescent intensity of the aptamer solution after reaction. Error bars represent the standard deviation ($n = 3$ independent experiments). Data are presented as mean values ± SD. **g** Bacterial viabilities of EcN, 2ApCB, 5ApCB, and 10ApCB by LB agar plate counting. Plates were incubated at 37 °C for 24 h prior to enumeration ($n = 3$ independent experiments). Data are presented as mean values ± SD. Significance was assessed using Student's *t* test (two-tailed). ns: no significance. **h** Degradation kinetics of the conjugated AS1411 in 90% phosphate-buffered serum solution at 37 °C. Error bars represent the standard deviation ($n = 3$ independent experiments). Source data are provided as a Source Data file.

incubation[33]. Extraordinarily, the stability of AS1411 grafted to the surface of EcN was greatly improved under the same experimental condition, with more than 70% of the conjugated aptamers remaining intact (Fig. 1h).

**Increased binding of ApCB with cancer cells.** CTL, a scrambled oligonucleotide with the same number of bases as AS1411 but without secondary structure, was used as a control. To examine the role of attached AS1411 on the invasion of EcN to cancer cells, cellular binding efficiencies of PBS, EcN$_{GFP}$, CTL$_{GFP}$, 2ApCB$_{GFP}$, 5ApCB$_{GFP}$, and 10ApCB$_{GFP}$ were assessed by using 4T1 cells, a cell line that overexpresses nucleolin (Supplementary Fig. 2a). LSCM imaging showed that the attachment of AS1411 to the cell surface could strengthen the binding of EcN with 4T1 cells (Fig. 2a). In accordance with the result of confocal imaging, flow cytometric analysis indicated that the binding efficiency increased with the grafting density of AS1411 increasing from $0.5 \times 10^5$ to $2.8 \times 10^5$ aptamers per cell, while decreased with further increasing the number to $5.5 \times 10^5$ (Fig. 2b). Under the same culture condition, the percentages of 4T1 cells bound with EcN after incubation with PBS, EcN$_{GFP}$, CTL$_{GFP}$, 2ApCB$_{GFP}$, 5ApCB$_{GFP}$, and 10ApCB$_{GFP}$ for 2 h were about 0.2%, 5.5%, 4.5%, 15%, 50%, and 30%, respectively (Supplementary Fig. 2b). These results demonstrated that decoration with aptamers could promote the interaction of bacteria with targeting cancer cells via specific ligand−receptor recognition, which depended on the grafting density.

To directly visualize the binding of ApCB with cancer cells, 4T1 cells after co-incubation with EcN and 5ApCB were observed by scanning electron microscopy (SEM), respectively (Fig. 3a). Representative SEM images showed that more bacteria were attached to the surface of 4T1 cells after culture with 5ApCB instead of EcN, which was consistent with the results of flow cytometric analysis. Flow cytometric analysis suggested that there was no significant difference in fluorescence intensity between the binding of 293T cells with EcN$_{GFP}$ and 5ApCB$_{GFP}$ (Fig. 3b, c). Meanwhile, LSCM images displayed that the binding efficiency of 5ApCB$_{GFP}$ with 293T cells was similar to that of EcN$_{GFP}$, which exhibited limited bacteria attaching onto the surface of 293T cells (Fig. 3d). Moreover, to further assess the binding efficiency of ApCB with cancer cells in vivo, a 4T1 tumor model was developed and the obtained tumor-bearing mice were intravenously dosed with PBS and $1 \times 10^7$ CFU of EcN or 5ApCB, respectively. Tumor tissues were sampled and stained with both FITC-labeled anti-*Escherichia coli* and 4′,6-diamidino-2-phenylindole (DAPI) at 12 h post injection (Fig. 3e).

The tumor-targeted colonization of ApCB was determined by tumor imaging using a 4T1 tumor-bearing mouse model. Based on the result of in vitro assay, 5ApCB with the highest binding efficiency was used to evaluate the tumor accumulation. To enable visual tracking, LuxCDABE engineered 5ApCB$_{Lux}$ were intravenously injected into the tumor-bearing mice through tail vein, while PBS, CTL$_{Lux}$, and EcN$_{Lux}$ without surface aptamers were administered as controls. At the indicated time points post

injection, the dosed mice were observed by in vivo imaging system (IVIS). As shown in Fig. 4a, the average luminescence signals in the tumors obtained by IVIS increased continuously with the time interval increasing from 12 to 60 h post administration. Importantly, at each time point, the average intensity of luminescence signals from the tumor site in mice injected with 5ApCB$_{Lux}$ was significantly higher than those of mice treated with PBS, CTL$_{Lux}$, and EcN$_{Lux}$, respectively (Fig. 4b). Particularly, at 60 h after injection, the relative signal intensity was 8-fold higher than those of the CTL$_{Lux}$ and EcN$_{Lux}$ groups. To quantify the tumor colonization, the biodistribution of the bacteria was examined at the endpoint of the animal study. After euthanasia, the heart, kidney, liver, lung, spleen, and tumor tissues were collected and then homogenized before spreading onto LB agar plates for bacterial counting. Accordingly, the bacterial number in the tumor tissue sectioned from mice injected with 5ApCB reached four times higher than those of the EcN and CTL groups (Fig. 4c). In contrast to the major organs, the distribution of bacteria in the tumor tissue was extraordinary, with around 99.9% of total bacteria colonizing inside the tumor. The overwhelming aggregation of EcN within the tumor tissue was ascribed to their preference for hypoxic, eutrophication, and the immunosuppressive microenvironment of tumors[41]. To prove this assumption, the localization of EcN within the 4T1 tumor at a relatively short period of time after injection was quantified by plate counting. As confirmed in Fig. 4d, the number of bacteria localized into the tumor at 12 h after dosing with 5ApCB achieved $1.75 \times 10^7$ CFU per gram tissue, which was 2-fold higher than that of unconjugated or CTL-decorated EcN. Namely, surface conjugation indeed offered a greater opportunity to localize tumor sites through a ligand−receptor interaction between aptamer-decorated bacteria and the tumor cells. In addition, routine blood and cytokine assays were performed to evaluate the potential side effects of 5ApCB. The values of white blood cell (WBC) and platelet (PLT) as well as the levels of interleukin-6 (IL-6) and interleukin-10 (IL-10) in mice dosed with 5ApCB were found to be similar to those of PBS group at 60 h post injection, implying the satisfied tolerance (Supplementary Fig. 3).

**Enhanced anticancer efficacy of ApCB.** In contrast to EcN, decoration with AS1411 resulted in a comparable increment in the binding of VNP with 4T1 cells (Supplementary Figs. 4, 5, and 6). Mouse subcutaneous tumor model of 4T1 was employed and the obtained tumor-bearing mice were randomly divided into three groups ($n = 5$) that were received with different treatments including PBS, VNP, and 5ApCB. It has been reported that a single intravenous injection of VNP at a dose ranging from $1 \times 10^4$ to $1 \times 10^6$ CFU per mouse is tolerable[42]. Thus, each mouse in VNP and 5ApCB groups was intravenously injected with $5 \times 10^5$ CFU of bacteria at day 0. A tumor inhibition study by concisely assessing tumor volume demonstrated that both VNP and 5ApCB could inhibit tumor growth compared to the PBS group (Fig. 5a, b). The variation of bodyweight recorded every other day after treatment

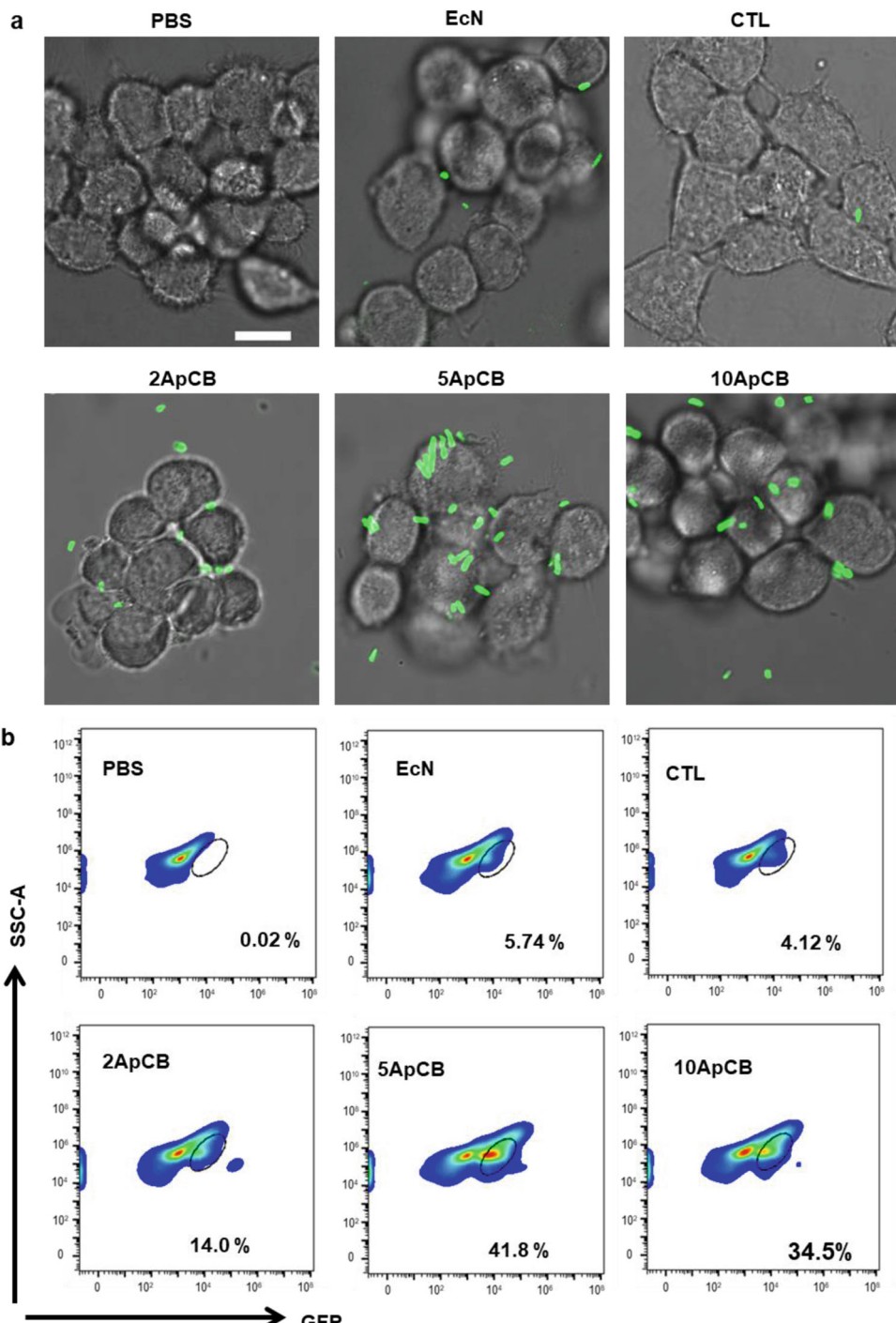

**Fig. 2 Binding of ApCB with 4T1 cells. a** Representative LSCM images of 4T1 cells after incubation with PBS, EcN_GFP, CTL_GFP, 2ApCB_GFP, 5ApCB_GFP, and 10ApCB_GFP at 37 °C for 2 h, respectively. Cells were rinsed with PBS before observation. Green channel means EcN producing GFP. Images are representative of three independent biological samples. Scale bar: 10 μm. **b** Flow cytometric analysis of the co-incubated 4T1 cells. Flow plots are representative of three independent biological samples. Source data are provided as a Source Data file.

revealed the limited side effects of the bacteria under the experimental dosage, as displayed in Fig. 5c.

To disclose the bacterial-mediated immune responses, the activation of T cells in the tumor tissue was analyzed at the end of the treatment. Nuclear nonhistone protein Ki67 was used as a marker to detect the infiltration of CD3$^+$ T cells in the tumor[43]. As shown in Fig. 5d, Supplementary Figs. 7a and 10, the proliferation of CD3$^+$ T cells in mice treated with 5ApCB presented an apparent increase in contrast to both PBS and VNP

groups. Moreover, treatment with 5ApCB significantly elevated the percentage of CD3$^+$CD4$^+$ T cells (Fig. 5e, Supplementary Figs. 7b and 10), which suggested that the aptamer-decorated bacteria could boost more efficient tumor infiltration of immunologic effector cells. The percentage of tumor cytokine interferon-γ (IFN-γ) and tumor necrosis factor-α (TNF-α), which predominantly regulates the cell-mediated immune responses[44], was further examined and the results showed a striking increase in mice treated with 5ApCB (Fig. 5f and g, Supplementary Figs. 7c

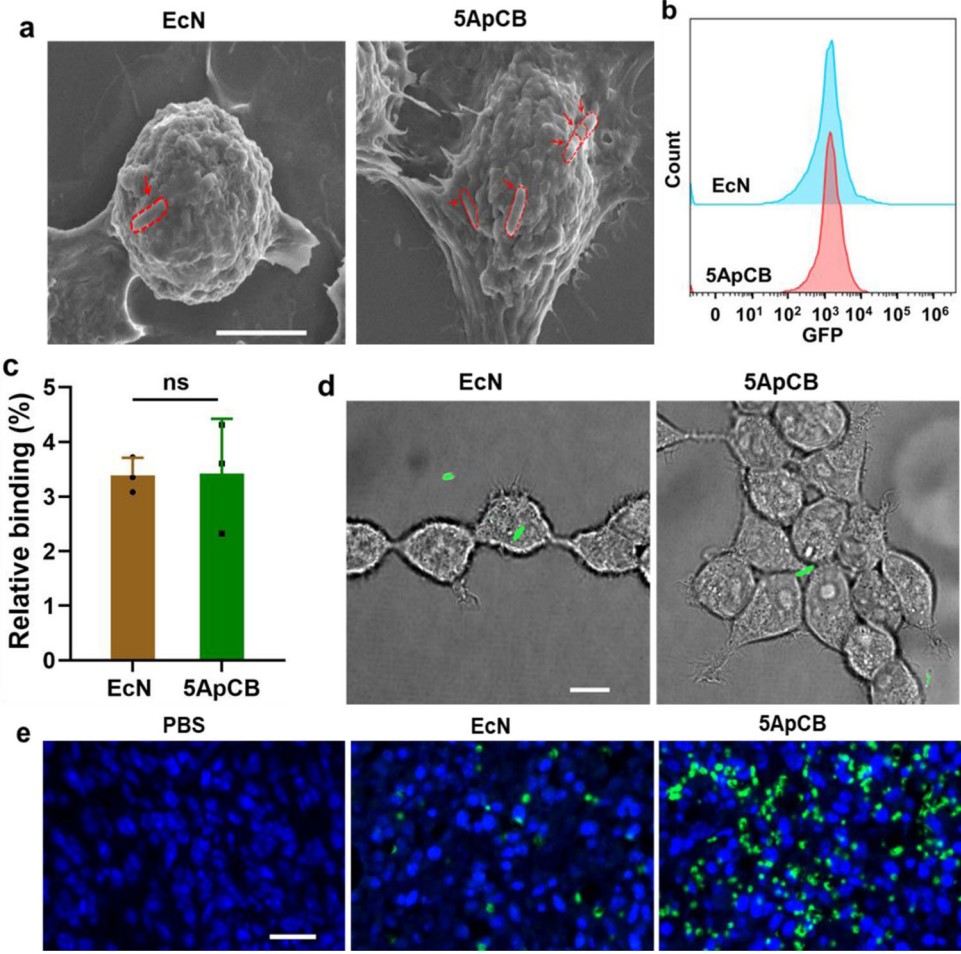

**Fig. 3 Specific binding of ApCB with cancer cells. a** Representative SEM images of 4T1 cells after incubation with EcN and 5ApCB at 37 °C for 2 h, respectively. Bacteria were circled by a red dotted line. Images are representative of three independent biological samples. Scale bar: 5 μm. **b** Flow cytometric analysis of 293T cells after co-incubation with EcN$_{GFP}$ and 5ApCB$_{GFP}$ at 37 °C for 2 h, respectively. **c** Percentages of 293T cells binding with EcN$_{GFP}$ and 5ApCB$_{GFP}$ after co-incubation at 37 °C for 2 h, respectively. Error bars represent the standard deviation ($n = 3$ independent experiments). Data are presented as mean values ± SD. Significance was assessed using Student's $t$ test (two-tailed). ns: no significance. **d** Typical LSCM images of 293T cells after incubation with EcN$_{GFP}$ and 5ApCB$_{GFP}$ at 37 °C for 2 h, respectively. Cells were rinsed with PBS before observation. Green channel means EcN producing GFP. Images are representative of four independent biological samples. Scale bar: 10 μm. **e** Confocal images of tumor tissues sectioned at 12 h after intravenous injection of $1 × 10^7$ CFU bacteria. Green and blue indicate EcN stained with FITC-labeled anti-*Escherichia coli* and nuclei stained with DAPI. Images are representative of three independent biological samples. Scale bar: 20 μm. Source data are provided as a Source Data file.

and 8a). The upgraded levels of TNF-α and IFN-γ in the tumor tissue indicated the activation of antitumor immune responses by 5ApCB treatment. Terminal deoxynucleotidyl transferase dUTP nick end labeling (TUNEL) staining depicted the highest level of tumor cell apoptosis induced by 5ApCB (Fig. 5h and Supplementary Fig. 8b). In addition, hematoxylin and eosin (H&E) staining exhibited prominent necrosis of the tumor tissues after treatment with the conjugated bacteria (Fig. 5i). Taken together, the inhibition of tumor growth could be attributed to the aptamer-mediated enhanced localization in the tumor site and the bacterial associated activation of antitumor immune responses.

**Versatility of ApCB**. TLS11a-based ApCB (T-ApCB) was synthesized similarly by co-incubating amino-functionalized TLS11a with VNP (Fig. 6a). Details for the preparation of T-ApCB were described in the Supplementary Information. Correspondingly, T-5ApCB was fabricated by incubation of $1 × 10^8$ CFU of VNP with 5 nmol of TLS11a. Flow cytometric analysis showed that H22 cells co-incubated with T-5ApCB$_{GFP}$ presented stronger fluorescence intensity than those cultivated with VNP$_{GFP}$,

demonstrating strengthened affinity of conjugated bacteria to bind with cancer cells (Fig. 6b and Supplementary Fig. 9). We next evaluated the antitumor efficacy of the resulting T-5ApCB in H22 tumor-bearing mice. After a single intravenous injection of PBS, $5 × 10^5$ CFU of VNP or T-5ApCB at day 0 (tumor size ~100 mm$^3$), both the tumor size and bodyweight of the treated mice were recorded. As plotted in Fig. 6c, d, T-5ApCB group appeared a significant inhibition of tumor growth in comparison with PBS and VNP groups, which could be explained by that surface conjugation of aptamers increased the intratumoral localization of bacteria. Survival rate of different groups further validated that treatment with T-5ApCB endowed H22 tumor-bearing mice with a significantly extended survival period, as reflected by that near 70% of the treated mice survived successfully in 36 days (Fig. 6e). While none of the mice dosed with PBS or VNP could survive under the same conditions. In addition, the variation of bodyweight during treatment revealed that administration with bacteria slightly dropped mouse bodyweight, which recovered rapidly to the same level of PBS group (Fig. 6f). The slight fluctuation in bodyweight implied the limited side effects of 5ApCB at the experimental dosage.

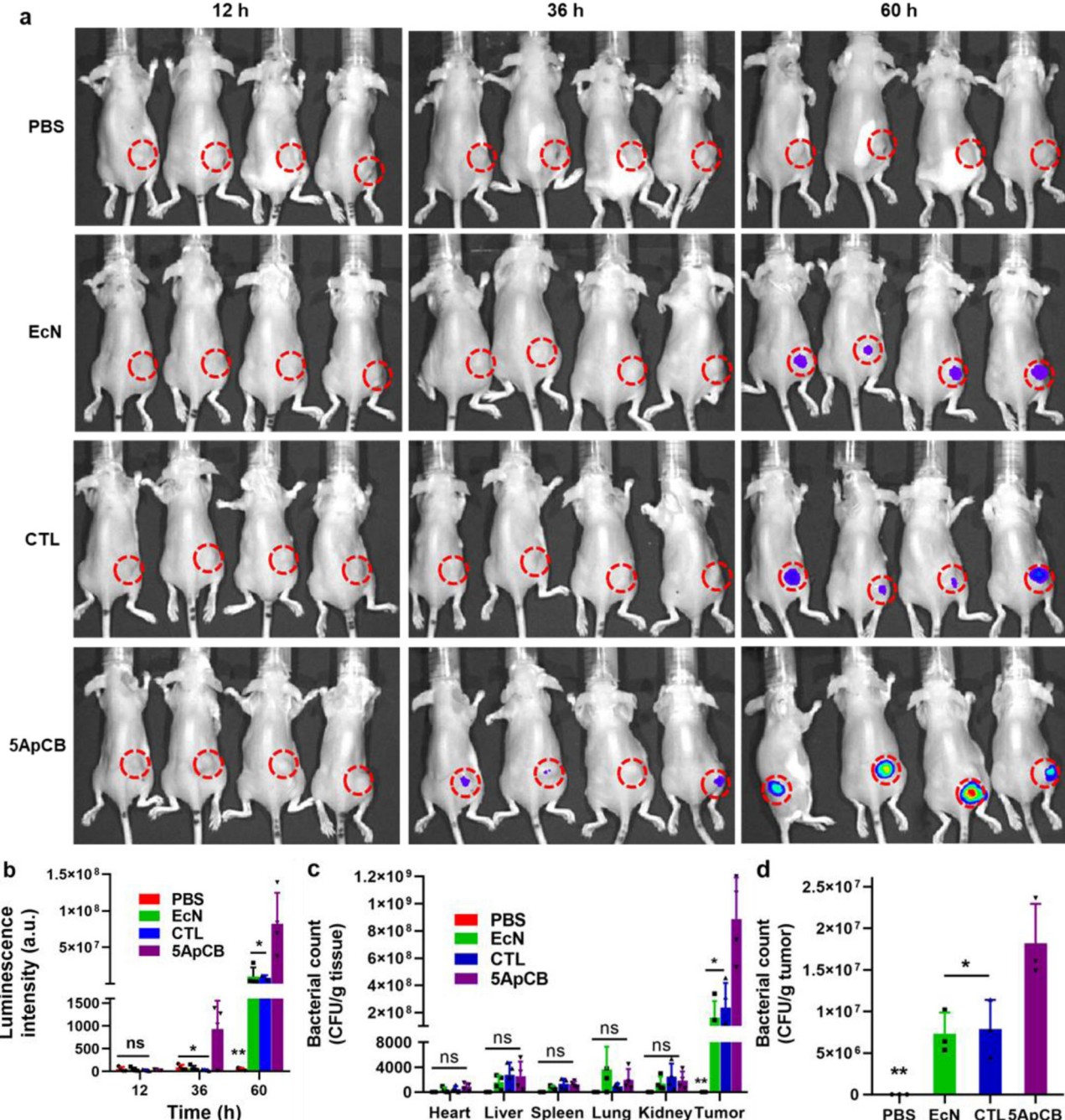

**Fig. 4 Biodistribution of ApCB in tumor-bearing mice. a** IVIS imaging of 4T1 tumor-bearing mice at 12, 36, and 60 h after tail vein injection with PBS or
$1 \times 10^7$ CFU EcN$_{Lux}$, CTL$_{Lux}$, and 5ApCB$_{Lux}$, respectively. **b** Average intensity of luminescence signals from the tumor site at 12, 36, and 60 h after injection.
Luminescence intensity (atomic units, a.u.) was quantified by a Caliper IVIS Lumina II system using a region-of-interest drawn to encompass the entire
solid tumor ($n = 4$). Data are presented as mean values ± SD. Significance was assessed using Student's $t$ test (two-tailed), giving $p$ values: 0.4115 for
5ApCB vs PBS, 0.5746 for 5ApCB vs EcN, 0.6742 for 5ApCB vs CTL at 12 h; 0.0370 for 5ApCB vs PBS, 0.0339 for 5ApCB vs EcN, 0.0287 for 5ApCB vs
CTL at 36 h;0.0081 for 5ApCB vs PBS, 0.0131 for 5ApCB vs EcN, 0.0132 for 5ApCB vs CTL at 60 h. ns: no significance. **c** Biodistribution of bacteria in the
major organs and tumor tissue of mice at 60 h after tail vein injection with PBS or $1 \times 10^7$ CFU of EcN, CTL, and 5ApCB, respectively. Numbers of bacteria
were quantified by plate counting. Data are presented as mean values ± SD ($n = 4$). Significance was assessed using Student's $t$ test (two-tailed), giving $p$
values: 0.0739 for 5ApCB vs PBS, 0.1990 for 5ApCB vs EcN, 0.1831 for 5ApCB vs CTL in heart; 0.1165 for 5ApCB vs PBS, 0.3731 for 5ApCB vs EcN, 0.9173
for 5ApCB vs CTL in liver; 0.0536 for 5ApCB vs PBS, 0.2886 for 5ApCB vs EcN, 0.8260 for 5ApCB vs CTL in spleen; 0.1069 for 5ApCB vs PBS, 0.4269 for
5ApCB vs EcN, 0.3103 for 5ApCB vs CTL in lung; 0.2162 for 5ApCB vs PBS, 0.8439 for 5ApCB vs EcN, 0.5253 for 5ApCB vs CTL in kidney; 0.0012 for
5ApCB vs PBS, 0.0138 for 5ApCB vs EcN, 0.0108 for 5ApCB vs CTL in tumor. ns: no significance. **d** Numbers of EcN colonized within the tumor at 12 h after
injection. Error bars represent the standard deviation ($n = 3$). Data are presented as mean values ± SD. Significance was assessed using Student's $t$ test
(two-tailed), giving $p$ values: 0.0027 for 5ApCB vs PBS, 0.0251 for 5ApCB vs EcN, 0.0390 for 5ApCB vs CTL. *$p < 0.05$, **$p < 0.01$. Source data are
provided as a Source Data file.

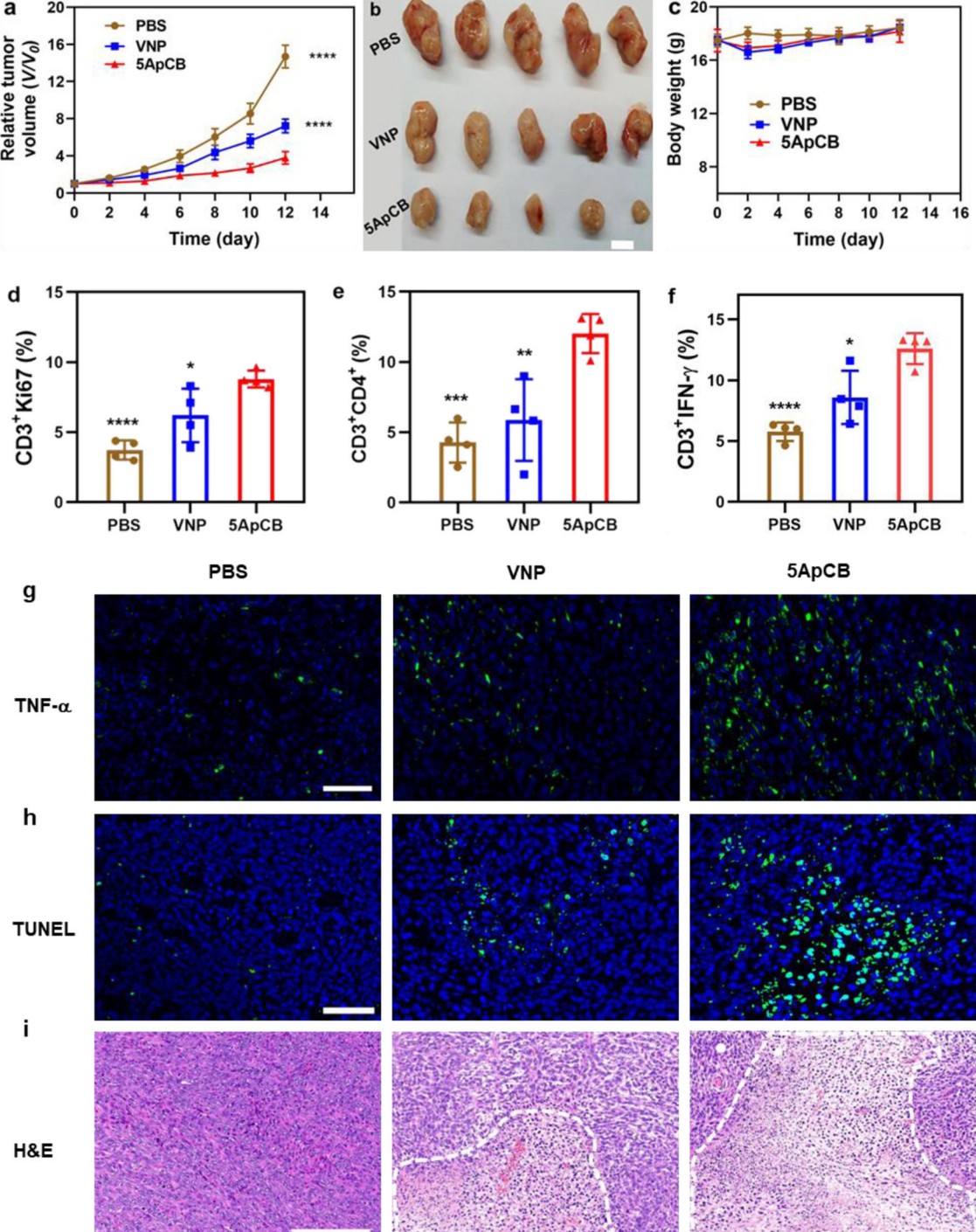

**Fig. 5 In vivo therapeutic efficacy of ApCB. a** Relative tumor growth after different treatments ($n = 5$). Data are presented as mean values ± SD. Significance was assessed using Student's *t* test (two-tailed), giving *p* values: 0.000005 for 5ApCB vs PBS; 0.00002 for 5ApCB vs VNP. **b** Representative photographs of the tumor tissues harvested from mice at the end of the treatment. Scale bar: 1 cm. **c** Variation of bodyweight during each treatment ($n = 5$). Data are presented as mean values ± SD. Percentage of **d** Ki67 (gated on CD3$^+$ cells) inside the tumors after treatment ($n = 4$). Data are presented as mean values ± SD. Significance was assessed using Student's *t* test (two-tailed), giving *p* values: 0.00003 for 5ApCB vs PBS; 0.0411 for 5ApCB vs VNP. **e** Flow cytometric analysis of the population of CD4$^+$ T cells (gated on CD3$^+$ cells) within the tumors after treatment ($n = 4$). Data are presented as mean values ± SD. Significance was assessed using Student's *t* test (two-tailed), giving *p* values: 0.0002 for 5ApCB vs PBS; 0.0088 for 5ApCB vs VNP. Percentage of **f** IFN-γ (gated on CD3$^+$ cells) inside the tumors after treatment ($n = 4$). Data are presented as mean values ± SD. Significance was assessed using Student's *t* test (two-tailed), giving *p* values: 0.00009 for 5ApCB vs PBS; 0.0192 for 5ApCB vs VNP. Images of tumor tissues stained with **g** TNF-α (green) and **h** TUNEL (green). Blue indicates nuclei stained with DAPI. Images are representative of three independent biological samples. Scale bar: 50 μm. **i** Typical H&E staining images of the sectioned tumors after different treatments. Necrotic areas were circled by a white dotted line. Images are representative of three independent biological samples. Scale bar: 200 μm. *$p < 0.05$, **$p < 0.01$, ***$p < 0.001$, ****$p < 0.0001$. Source data are provided as a Source Data file.

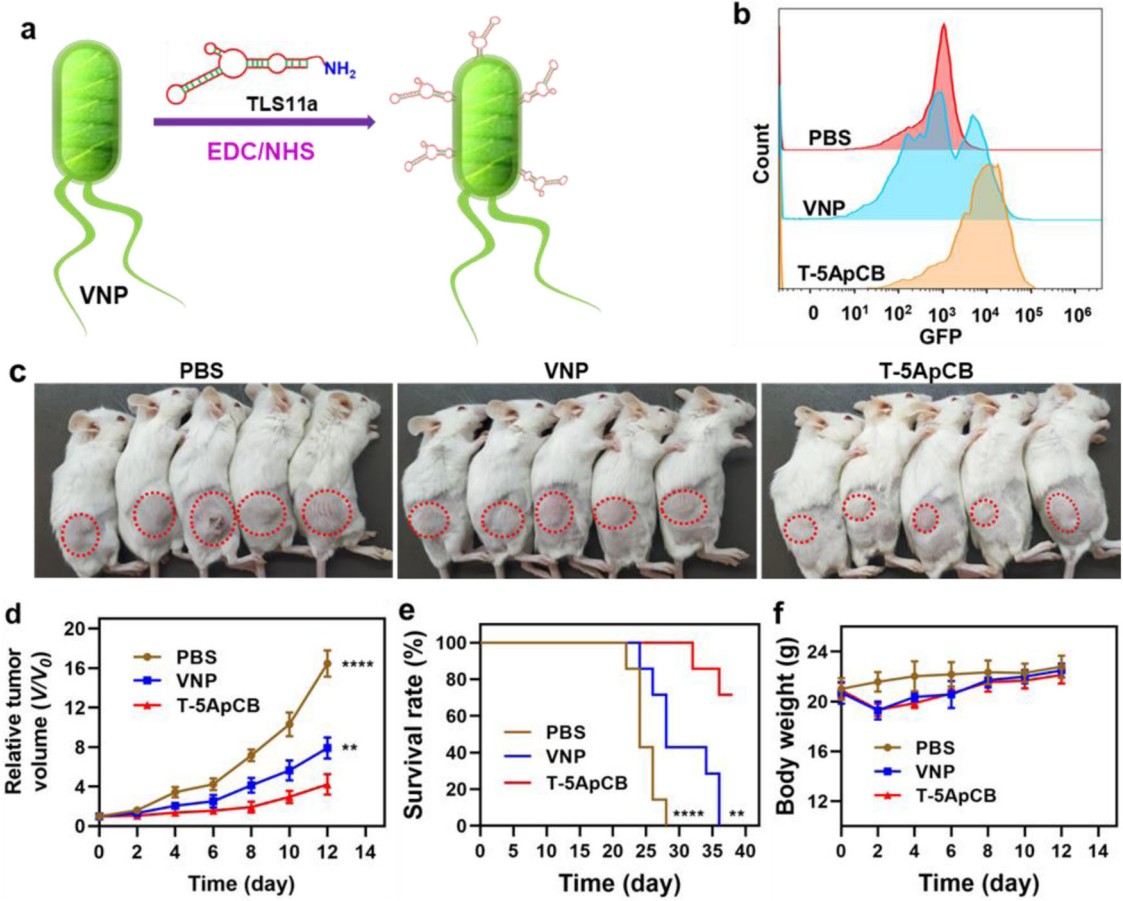

**Fig. 6 Implementation of ApCB with aptamer TLS11a in an H22 tumor-bearing model. a** Preparation of ApCB by conjugating VNP with TLS11a via amide condensation. **b** Flow cytometric analysis of H22 cells after co-incubation with PBS, VNP, and T-5ApCB respectively at 37 °C for 1 h. Flow plots are representative of three independent biological samples. **c** Digital photos of mice at day 12 after treatment. Tumors were marked in red circles. H22 tumor-bearing mice (inoculated with $1 \times 10^6$ carcinoma cells) were randomly divided into three groups and intravenously injected with PBS and $5 \times 10^5$ CFU of VNP or T-5ApCB upon tumor size reaching ~100 mm³ (defined as day 0). **d** Relative tumor growth after different treatments ($n = 5$). Data are presented as mean values ± SD. Significance was assessed using Student's $t$ test (two-tailed), giving $p$ values: 0.000002 for T-5ApCB vs PBS; 0.0052 for T-5ApCB vs VNP. **e** Survival curves of H22 tumor-bearing mice after different treatments ($n = 7$). Data are presented as mean values ± SD. Significance was assessed using Student's $t$ test (two-tailed), giving $p$ values: $p < 0.0001$ for T-5ApCB vs PBS; 0.0017 for T-5ApCB vs VNP. **f** Variation of bodyweight during treatment. ** $p < 0.01$, ****$p < 0.0001$. Source data are provided as a Source Data file.

## Discussion

In view of the specificity to bind with nucleolin overexpressing on the membrane of various cancer cells[45], we selected aptamer AS1411 to prepare ApCB. On the other hand, as a well-known probiotic bacterium, EcN was chosen as a model strain due to its ability to colonize in different tumors[46]. Both LSCM images and flow cytometric analysis indicated that bacteria were conjugated with Cy5-labeled AS1411 by amide condensation. Given that the surface density of aptamers could influence the binding ability of the modified bacteria[47], we then investigated whether the number of grafted aptamers could be tuned by varying the feed ratio. The results suggested a concentration-dependent conjugation, which meant that the grafting density of AS1411 could be controlled by changing the feed ratio. The number of aptamers decorated on bacteria was quantified by calculating the difference of fluorescent intensity of the aptamer solution after the reaction. In good agreement with the result of flow cytometric analysis, the number of attached aptamers increased with the incubation concentration of AS1411. Both the preparation procedure and the conjugated aptamers displayed negligible toxic side effects on bacterial viability. Meanwhile, the conjugation could strengthen the resistance of aptamers against serum-mediated degradation, suggesting the

potential of ApCB to be utilized in vivo. With the guidance of surface-anchored aptamers, we expected that ApCB could present specific affinity to cancer cells overexpressing the corresponding receptors. Interestingly, incubation with 5ApCB<sub>GFP</sub> visualized the highest number of bound EcN on the cell surface. Although there was much more grafted AS1411 on EcN surface, 10ApCB<sub>GFP</sub> bound less efficiently than 5ApCB<sub>GFP</sub>. This might be ascribed to the influence of grafting density, such as the accessibility of adjacent aptamers as well as the steric effects of neighboring aptamers on the surface[48]. To verify the specific binding with cancer cells, EcN and 5ApCB were separately added to the culture medium of 293T cells, a cell line that presents low nucleolin expression[49,50]. The results indicated that the attachment of AS1411 onto bacteria played a critical role for ApCB to specifically bind with cancer cells. Moreover, mice injected with 5ApCB presented a largely increased number of bacteria binding with cancer cells compared to EcN, further demonstrating the specific and enhanced binding of bacteria by conjugation with aptamers on the surface. Having confirmed the presence of exogenous function that could specifically bind with tumor cells after conjugation, we next turned our attention to evaluate the ability of ApCB to target the tumor site. In comparison to undecorated or

CTL-decorated EcN, the notably enhanced accumulation within the tumor tissue after injection with 5ApCB was attributed to the improved localization of bacteria following administration, which was resulted from the specific binding with tumor cells.

To validate the versatility of our approach, we translated the surface conjugation of aptamers to VNP, a well-known strain that has been widely used for tumor biotherapy due to its high immunogenicity[51]. VNP was similarly conjugated with AS1411 and the surface density of aptamers was optimized by varying feed ratios. We found that the 5ApCB group showed significantly enhanced antitumor efficacy in comparison with the VNP group, suggesting that the increased tumor localization enabled by surface conjugation of aptamers played an important role in tumor biotherapy. Furthermore, although the bodyweight of mice dropped slightly after injection with the bacteria, the mice in both the VNP and 5ApCB groups recovered rapidly as reflected by the return of the bodyweight to the same level of the PBS group. It has been reported that the colonization of *Salmonella* can provoke the immunosuppressive microenvironment of tumors and elicit antitumor immunity[52]. Our data suggested that treatment with 5ApCB significantly elevated the cell-mediated immune responses in contrast to both the PBS and VNP groups. To explore the generality of this strategy as a versatile platform for different aptamers, TLS11a, an aptamer that is identified in 2008 using a modified whole-cell SELEX (systematic evolution of ligands by exponential enrichment) procedure and shows a high binding affinity to a given hepatocellular carcinoma cell line, was used for further investigation[53]. The corresponding tumor model of H22 hepatocellular carcinoma was developed to evaluate the effectiveness of conjugating aptamers to bacterial surface[54,55]. The significant inhibition of tumor growth in the T-5ApCB group proposed the universality of this approach to prepare diverse ApCB and verified the ability of ApCB to increase the localization of conjugated bacteria in solid tumors for enhanced biotherapy. The levels of inflammatory reactions caused by bacteria were similar to those of the PBS group at 60 h post injection. Indeed, several clinical trials of bacteria-mediated tumor therapy have completed or are currently ongoing[56–60], implying the safety of bacteria-mediated tumor therapy.

In summary, we have reported a strategy of conjugating aptamers to bacterial surface for endowing bacteria with tumor-specific accumulation following administration. The conjugation is based on a single-step and cytocompatible amidation procedure, which is implementable to diverse Gram-negative bacteria. The grafting density of aptamers on the bacterial surface can be simply controlled by tuning the feed ratio and the attachment is capable to increase the resistance of anchored aptamers against degradation due to the increase of steric hindrance towards the enzymatic hydrolysis by nucleases. With an optimized number of $2.8 \times 10^5$ aptamers per cell, the conjugated bacteria exhibit the highest specific affinity with tumor cells in vitro. Strikingly, the optimal bacteria reached near 2- and 4-fold higher colonization in tumor tissue at 12 and 60 h after systemic administration in vivo. The therapeutic value in tumor biotherapy has been validated by injection with aptamer-conjugated attenuated *Salmonella* in both 4T1 and H22 tumor-bearing mice. As expected, the decorated bacteria display significantly enhanced antitumor efficacy, with effective infiltration of immunologic effector cells and expression of tumor cytokines inside the tumor. This work discloses how bacterial bioactivities can be manipulated by surface conjugation with aptamers. With the potential to address the low efficacy associated with bacterial-based therapy, we anticipate the application of this strategy as a versatile platform to develop various bacterial therapeutics for enhanced tumor treatment.

## Methods

**Materials and strains**. 1-Ethyl-3(3-dimethylaminopropyl) carbodiimide (EDC) and *N*-hydroxysuccinimide (NHS) were purchased from Adamas-beta (Shanghai, China). Roswell Park Memorial Institute (RPMI) 1640 medium antibiotic/anti-mycotic solution and phosphate-buffered saline (1× PBS) were provided by Sigma-Aldrich (USA). AS1411 (5′-GGT GGT GGT GGT TGT GGT GGT GGT GGT TTT TTT TTT TT-NH₂-3′), TLS11a (ACA GCA TCC CCA TGT GAA CAA TCG CAT TGT GAT TGT TAC GGT TTC CGC CTC ATG GAC GTG CTG TTT-NH₂-3′), and CTL (5′-CCT CCT CCT CCT TCT CCT CCT CCT CCT TTT TTT TTT T T-NH₂-3′) were synthesized by Tsingke Biological Technology Co., Ltd (Shanghai, China). EcN and *Salmonella typhimurium* were purchased from China General Microbiological Culture Collection Center (GMCC, China). 4T1 mammary gland carcinoma cell line and 293T cell line were obtained from American Type Culture Collection (ATCC, CRL-2539; CRL-3216) and H22 hepatocellular carcinoma cell line was purchased from BeNa Culture Collection (BNCC) (catalog: BNCC338327). Both cell lines were cultured in Dulbecco's modified Eagle medium (Sigma, USA) supplemented with 10% (v/v) inactivated fetal bovine serum (FBS) (Sigma, USA) and 1% (v/v) antibiotic/anti-mycotic solution (Sigma, USA) at 37 °C in an incubator with 5% CO₂. Regular mycoplasma evaluations were performed of the cell culture environment to ensure the absence of mycoplasma contamination. Plasmids pBBR1MCS2-Tac-GFP, pMD18-luxCDABE and all other reagents were purchased from domestic suppliers and used as received.

**Animals**. BALB/c female mice (6−8 weeks, 18−20 g) were provided by Jiesijie laboratory animal center (Shanghai, China). All mice were housed and fed under 12/12 h dark/light cycle and specific pathogen-free (SPF) conditions (ambient temperature (25 °C) and humidity (55%). The protocol of the animal study was reviewed and approved by the Institutional Animal Care and Use Committee of Shanghai Jiao Tong University School of Medicine. Mice were euthanized when the tumor size reached more than 2000 mm³. The maximal tumor size/burden was not exceeded.

**Characterization**. Photoluminescence (PL) spectra were measured on a FluoroMax-4 spectro-fluorometer. The fluorescence images of cells were characterized by confocal laser scanning microscopy (CLSM, Leica, SP8). Flow cytometric (Beckman CytoFlex) analysis was conducted for quantitative detection of cellular fluorescence. In vivo imaging of mice was performed by in vivo imaging system (Caliper LifeSciences, USA). Morphology observation was performed on a scanning electron microscopy (SEM) (Sirion 200, USA).

**Growth of bacteria**. EcN carrying pBBR1MCS2-Tac-GFP, or pMD18-luxCDABE were grown at 37 °C overnight in 10 mL liquid LB medium with a supplement of 50 μg/mL gentamicin or kanamycin. *Salmonella typhimurium* was grown at 37 °C in tryptic soy broth (TSB) medium. Overnight culture was diluted 1:50 (v/v) to fresh LB liquid medium and grown at 37 °C for 2−3 hours. Bacteria were collected by centrifugation at 6000 × g for 1 min and resuspended in ice-cold PBS. Bacterial counts were determined by making dilutions of bacterial suspension, culturing them on LB agar plates at 37 °C overnight and counting the colony-forming units (CFU).

**Preparation of ApCB**. Amino-functionalized aptamer was linked on the surface of bacteria by amide condensation. Briefly, $1 \times 10^8$ of bacterial cells were dispersed in 1 mL PBS solution, and 0.1 mL of amino-functionalized aptamer (0.05 μmol/mL) was added into the bacterial suspension together with 0.55 mg EDC and 0.65 mg NHS at room temperature. In amide condensation, EDC was able to activate the carboxyl group forming an unstable *O*-acylisourea. Adding NHS to the reaction mixture could stabilize the activated carboxyl groups and subsequently improve the conversion of amide condensation. After stirring for 3 h, the modified bacteria were separated by centrifugation (6000 × g, 1 min) and washed with PBS three times.

**Calculation of the average number of aptamers per bacterial**. First, several Cy5-labeled AS1411 solutions with certain concentrations were prepared. Then, a standard curve was established by plotting fluorescence intensity against Cy5-labeled AS1411 concentrations with the help of photoluminescence spectroscopy using excitation at 647 nm and emission at 670 nm. After determining the regression equation for the standard curve, the concentrations of free Cy5-labeled AS1411 could be calculated after the reaction. The average number of aptamers per bacterial cell was calculated as follows:

$$N_{avg} = ((c_0 v - c_a v) \times NA)/N_a$$

where $N_{avg}$ is the average number of aptamers per bacterial cell, $c_0$ is the concentration of free Cy5-labeled AS1411 before reaction, $c_a$ is the concentration of free Cy5-labeled AS1411 after reaction, $v$ is the volume that the reactions took place in, $NA$ is the Avogadro constant $6.02 \times 10^{23}$, $N_a$ is the total number of bacterial cells.

**Serum stability**. To investigate serum stability, $1 \times 10^9$ CFU ApCB with Cy5-labeled AS1411 were suspended in 5 mL PBS supplemented with 10% FBS at 37 °C for predetermined time points including 0, 1, 4, 8, 12, 24, and 48 h. At the end of

each time point, 100 µL of each sample was withdrawn and centrifuged. After rinsing with PBS, the final samples were resuspended in 100 µL PBS and placed in quartz cuvettes. The fluorescence emission intensity was measured using excitation at 647 nm and emission at 670 nm at room temperature. The concentrations of conjugated aptamers were calculated by using the regression equation obtained from the standard curve.

**Cellular uptake**. Cells were seeded at a density of $2 \times 10^5$ cells per confocal dish in 1 mL RPMI-1640 medium with 10% FBS and antibiotics. 4T1 cells were incubated for 24 h at 37 °C in 5% $CO_2$. Then, the medium was replaced by fresh 1640 medium and treated with equal volume of EcN or ApCB at 37 °C for 2 h. Subsequently, after being washed with PBS for three times, the cells were viewed using LSCM. For quantitative analysis of the uptake behavior, the cells were digested and collected for flow cytometric analysis. To directly observe the binding of bacteria with cancer cells, the co-incubated cells were fixed with 4% glutaraldehyde in 0.5 mL of PBS at 4 °C for 40 min. Next, the cells were dehydrated in a series of ethanol/water solution with increasing ethanol content from 30 to 100%. Finally, the associated ethanol in the cells was removed by freeze-drying.

**Subcutaneous tumor model**. The experiments were performed on female BALB/C mice (6−8 weeks) bred under specific pathogen-free (SPF) conditions for 6 days. For both 4T1 and H22 tumor models, $1 \times 10^6$ cells in 100 µL of serum-free 1640 medium were injected subcutaneously into the right hind leg of each BALB/C mouse. The tumor volume of each mouse was measured by caliper and calculated using the formula: $(width)^2 \times length \times 0.5$.

**In vivo binding of bacteria with cancer cells**. To develop a 4T1 tumor-bearing mouse model, $1 \times 10^6$ cells in 100 µL of serum-free 1640 medium were injected into the right hind leg of each Balb/c nude mouse. The inoculated mice were grouped randomly into three groups, including PBS, EcN ($1 \times 10^7$ CFU), and 5ApCB ($1 \times 10^7$ CFU) after the size of tumors reached ~200 cm³. A volume of 100 µL of each above-mentioned solution was administered through the tail vein. For bacterial staining, mice were euthanized to sample the tumor tissues at day 12 post injection. The samples were fixed in 4% paraformaldehyde fixative and then transferred to sucrose-containing PBS. After overnight immersion at 4 °C, the samples were cut into slides and incubated with FITC-labeled anti-*Escherichia coli* antibody in PBS containing 0.1% normal serum for 1−2 h at room temperature in dark. Cell nuclei were stained by 4′,6-diamidino-2-phenylindole (DAPI). After washing with PBS for three times, the samples were mounted for imaging.

**Biodistribution**. To study the in vivo biodistribution of EcN and 5ApCB, the mice bearing 4T1 tumor were injected via the tail vein with 0.1 mL of normal saline containing $1 \times 10^7$ CFU of EcN or 5ApCB. Four mice were used for each group. All the mice were imaged by in vivo imaging system at the indicated time points and then sacrificed for tissue collection. The liver, spleen, lung, heart, kidney, and tumor were sampled and homogenized in a glass homogenizer. Equal weight of each homogenate was diluted serially with LB and 50 µL of each dilution was spread onto LB agar plates with antibiotics before overnight incubation at 37 °C for bacterial counting.

**Tumor imaging**. To develop a 4T1 tumor-bearing mouse model, $1 \times 10^6$ cells in 100 µL of serum-free 1640 medium were injected into the right hind leg of each Balb/c nude mouse. The inoculated mice were grouped randomly after the tumors reached ~200 cm³ ($n = 4$ for each group). To examine the accumulation of EcN and 5ApCB in the tumor, the tumor-bearing mice were intravenously injected with $1 \times 10^7$ CFU of EcN or 5ApCB expressing LuxCDABE. Fluorescence imaging was carried out at 12, 48, and 60 h post injection. All the mice were imaged by in vivo imaging system and sacrificed at the predetermined time points for further analysis. All images were analyzed by Caliper IVIS Lumina II system. In each group, region-of-interests (ROIs) were defined for the tumor tissue. The average intensity of tumor ROIs was calculated and the intensity of background within this ROI was subtracted for each image frame.

**In vivo bacterial-mediated cancer therapy and histological analysis**. Tumor-bearing mice were divided into three groups after the size of the tumors reaching ~100 cm³ ($n = 5$ for each group), including PBS, VNP ($5 \times 10^5$ CFU) and 5ApCB or T-5ApCB ($5 \times 10^5$ CFU). 100 µL of each above-mentioned solution was administered through the tail vein. The size of tumor was measured every other day with a digital caliper. For histology analysis, the tumors were harvested, fixed in 4% paraformaldehyde solution, embedded in paraffin, and then stained with hematoxylin and eosin (H&E). Meanwhile, the tumor tissues of each group were collected for immunofluorescence staining analysis of TUNEL and TNF-α.

**Flow cytometric assay for immune responses**. Tumor tissues were harvested and treated with 1 mg/mL collagenase I (Gibco) for 1 h and ground using the rubber end of a syringe. Cells were filtered through nylon mesh filters and washed with PBS. Cells were further stained with corresponding fluorochrome-conjugated antibodies. To analyze the T-cell subsets in draining tumor, single-cell suspensions

prepared from these samples were examined by flow cytometry. The following primary antibodies were used: PerCP-Cyanine5.5-labeled anti-CD3 (BioLegend, catalog: 100328), APC-labeled anti-CD4 (BioLegend, catalog: 100412), PE-Cy7-labeled anti-IFN-γ (BioLegend, catalog: 505826), and FITC-labeled anti-Ki67 (eBioscience, catalog: 11-5698-82). Antibodies were used at a dilution of 1:200. Data analysis was acquired on a CytoFLEX and analyzed using CytExpert (Beckman Coulter, USA) and FlowJo (TreeStar, USA) software.

**Statistics and reproducibility**. All data were presented as means ± standard deviation (SD) from at least three independent runs. Statistical analysis was performed using Prism 8.0 (GraphPad, USA). Unpaired two-tailed Student's $t$ test (two-tailed) was used for comparison between two groups. In all cases, differences were considered statistically significant if $p < 0.05$ (*$p < 0.05$, **$p < 0.01$, ***$p < 0.001$, and ****$p < 0.0001$).

**Reporting summary**. Further information on research design is available in the Nature Research Reporting Summary linked to this article.

## Data availability
All data are available within the article or Supplementary information. Source data are provided with this paper.

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

## Acknowledgements

This work is supported by the National Key Research and Development Program of China (SQ2021YFA090162, J.L.), and the National Natural Science Foundation of China (21875135, J.L.; 32101218, Z.C.; 52101289, R.L.).

## Author contributions

J.L. and W.T. supervised the project. J.L. conceived and designed the experiments with Z.G. Z.G., Z.C., R.L. and K.L. performed all experiments. All authors analyzed and discussed the data. Z.G. and J.L. wrote the paper.

## Competing interests

The authors declare no competing interests.
