## [Peer Review File · Nature Communications]

Aptamer-assisted tumor localization of bacteria for enhanced biotherapyREVIEWER COMMENTS

Reviewer #1 (Remarks to the Author):

In this manuscript, the authors developed a simple and biocompatible amidation method to decorate bacteria with aptamers. By conjugating aptamers AS1411 to bacterial surface and optimizing the ratio, the authors achieved an enhanced tumor accumulation and anti-tumor activity. After conjugated with an average of 2.8×10^5 aptamers per cell, the bacteria can generate near 2- and 5-times higher accumulation in tumor tissue after 36 and 60 hours compared to unmodified bacteria. Overall, this is a promising strategy to promote the clinical transformation of biotherapies. The major conclusions are fully supported by the experimental data. I would suggest its publication on Nature communication after addressing the following issues.

1. The nucleolin expression and intracellular distribution of 4T1, H22 and 293T cells should be measured and confirmed by western blotting or flow cytometry.
2. In the abstract, the authors claimed "their implementation has been restricted by severe side effects". Therefore, I suggest the authors should carry out extensive safety evaluation experiments in detail.
3. In the introduction, the authors described "which separately achieves 2- and 5-fold higher colonization in tumor tissue at 12 and 60 hours post tail vein injection in mice". However, the result in figure 4b showed higher tumor accumulation at 36 h and 60 h. Please double check it.
4. Why the bacteria count is higher in EcN group than 5ApCB group in figure 4d.
5. Please provide the quantitative analysis of figure 6b.
6. In the schematic diagram, the "cancer cell" should be "cancer cells".
7. The "Figure 4b" in the manuscript is missing.
8. Please give out how the fluorescence intensity was quantified in figure 4c by in vivo data.

Reviewer #2 (Remarks to the Author):

In this work, Geng et al reported the conjugation of aptamers to bacterial surface by a simple and cytocompatible amidation procedure, which could improve the localization of bacteria in tumor site following tail vein injection. The authors found that the conjugation of aptamer to the surface of bacteria increased its stability in serum. By varying feed ratio, the optimized bacteria conjugated with a certain number of aptamers per cell showed the highest specificity to tumor cells. After systemic injection, the conjugated bacteria generated a few times higher accumulation in tumor site in contrast to unmodified bacteria. The authors further implemented this approach with attenuated Salmonella, a strain that has entered phase I clinical trial. In both 4T1 and H22 tumor-bearing mice, aptamer-conjugated attenuated Salmonella presented significantly enhanced treatment efficacy, along with highly activated intratumoral immune responses. Overall, the conjugation of living bacteria with aptamers to improve their localization in tumor site is very innovative, and the conclusions are well supported by the in vitro and in vivo data. The findings would be of great interest to researchers in the field of drug delivery and cancer therapy. Therefore, the reviewer recommends the publication of this manuscript in Nature Communications provided the below minor issues have been appropriately addressed.

- 1) Compared to 4T1 cells, EcN and 5ApCB were separately added to the culture medium of 293T cells and the results showed no significant difference in fluorescence intensity between the binding of 293T cells with EcN and 5ApCB. The explanation of the mechanism was missing and should be added.
- 4) The authors mentioned in the manuscript that the mice injected with 5ApCB showed the strongest fluorescence signal, which could extend to 60 hours post injection. However, the authors did not describe the details on how to capture the fluorescence in tumor imaging section, as which is critical for the assessment of targeting effect.
- 3) In both 4T1 and H22 tumor-bearing mouse models, the bodyweight of mice dropped in the first 2 days post-injection. Was this caused by the injection of too much bacteria? How to decide the dose of bacteria? These should be clarified.
- 4) In Figure 5g, it was claimed that histology images of the sampled tumor tissues after treatment with conjugated bacteria suggested the largest damaged area. The damage areas need to be pointed out in the H&E images and the description should be added to the figure legend.
- 5) Figure 6b included the flow cytometric analysis data of H22 cells after co-incubation with PBS,

VNP, and T-5ApCB at 37 °C for 1 hour. The geometric mean fluorescence intensity of H22 cells after co-incubation with PBS, VNP, and T-5ApCB should be added.

6) In the section of enhanced anticancer efficacy of ApCB: VNP were similarly conjugated with AS1411 and the antitumor efficacy of the resulting 5ApCB was assessed in vivo. Although 5ApCB showed significantly enhanced antitumor efficacy in comparison with VNP group, the binding efficiency of VNP and 5ApCB with 4T1 cells should be examined in vitro.

7) The legends of Figure 5 should be consistent with its contents. Images of tumor tissues stained with (g) TNF- α and (h) TUNEL. Scale bar: 50 μ m. (i) Typical H&E staining images of the sectioned tumors after different treatments. Scale bar: 200 μ m.

Reviewer #3 (Remarks to the Author):

In the research article, Aptamer-assisted tumor localization of bacteria for enhanced biotherapy, the authors demonstrated the design, synthesis, and efficacy of an anticancer biotherapy based on non-virulent gram-negative bacterial strains that have been chemically modified by the conjugation with DNA aptamers to the cell surface. The authors showed that this system has increased efficacy against in vivo cancer models than equivalent, non-targeted, treatments. Additionally, the authors showed that the density of aptamers on the cell surface required optimization, as too many aptamers led to a reduced binding efficiency by the conjugated aptamers. While the work of some interest, the main issues that this article is facing is a lack of control experiments, a need to expand on their calculations in some areas, and a general need for the language to be cleaned up in some areas.

Major

Manuscript requires extensive review as grammatical errors can interfere with reader comprehension.

Scrambled sequences of the same lengths as aptamers should be used in all experiments to assess the effect of nucleic acid modifications.

Please clarify the calculations/equations used to determine the number of aptamers/cell from spectrofluorimetry results and flow cytometry.

It would be highly beneficial to the manuscript to provide individual names for each treatment so that they can be clearly distinguished from each other and the effects being described can be easily identified. For example, on page 11, when describing IVIS imaging, it is unclear what is fluorescing in the system as written.

The data from the PBS treatments needs to be shown in all cases. The same procedures which were followed for the other two treatments (non-conjugated and aptamer-conjugated) need to be followed and the results shown for the PBS treatment. For example, the bacterial culture counting plates for bacteria isolated from tumor tissues post-treatment in mice should be shown. This is an issue in all experiments comprising these three groups (PBS, Bacteria, Aptamer-Bacteria).

It would be beneficial to repeat the studies demonstrating the optimal surface density of the aptamers for each bacterial strain tested, instead of just the EcN.

Additional background information regarding the TLS11a aptamer (1-2 sentences) would be helpful.

Is there evidence that the amine-Aptamer only interacts with the cell walls or will this system react and covalently bond to any exposed carboxylic acid groups (e.g. on glutamate or aspartate residues as well).

Figure 4. The color scheme flips between each image. Please keep this consistent between panels. Additionally, the caption does not match for figure 4. This is also true for Figure 5.

Please add, if possible, the data from the PBS treated mice in this and all subsequent figures.

Preparation of ApCB-Please clarify the volume that the reactions took place in and in the corresponding discussion section, please add a brief statement on the role of EDC and NHS in the reaction.

Serum Stability-Please include more details regarding the experiment assessing serum stability. Cy5 should fluoresce regardless of the degradation status of the aptamer, so please clarify how the assay as described demonstrates the serum stability of the aptamer.

Minor

PBS is used interchangeably to refer to phosphate-buffered saline as well as phosphate-buffered serum

The "severe side effects" of bacterial-mediated biotherapies are not mentioned but in Figure 1b, TNF and INF are illustrated. Please elaborate on these side effects and the significance of the illustrated cytokines.

Pg. 10: Authors discuss the intravenous dosing of VNP, yet the corresponding confocal images are that of EcN

Reviewer: 1

In this manuscript, the authors developed a simple and biocompatible amidation method to decorate bacteria with aptamers. By conjugating aptamers AS1411 to bacterial surface and optimizing the ratio, the authors achieved an enhanced tumor accumulation and anti-tumor activity. After conjugated with an average of 2.8×10^5 aptamers per cell, the bacteria can generate near 2- and 5-times higher accumulation in tumor tissue after 36 and 60 hours compared to unmodified bacteria. Overall, this is a promising strategy to promote the clinical transformation of biotherapies. The major conclusions are fully supported by the experimental data. I would suggest its publication on Nature communication after addressing the following issues.

Response: We thank the reviewer very much for providing constructive comments on how to further improve the quality of our manuscript.

1. The nucleolin expression and intracellular distribution of 4T1, H22 and 293T cells should be measured and confirmed by western blotting or flow cytometry.

Response: Nucleolin is one of the major abundant proteins in the nucleus and also overexpressed on the surface of a number of cancer cell lines. According to the reviewer's suggestion, the expression of nucleolin on 293T, H22 and 4T1 cell surface was measured by flow cytometry (Figure S2a). The results indicated that the level of nucleolin expression on 4T1 cells was clearly higher than those of H22 and 293T cells.

2. In the abstract, the authors claimed "their implementation has been restricted by severe side effects". Therefore, I suggest the authors should carry out extensive safety evaluation experiments in detail.

Response: According to the reviewer's suggestion, the routine blood and cytokine assays were performed to evaluate the inflammatory reactions caused by bacteria. The values of white blood cell (WBC) and platelet (PLT) as well as the levels of interleukin-6 (IL-6) and interleukin-10 (IL-10) in mice dosed with 5ApCB were found to be similar to those of PBS group at 60 h post-injection (Figure S3), which was consistent with our previous results (Nat. Commun. 2019, 10, 3452). Actually, in phase I clinical trial of VNP20009, the obtained anticancer efficacy was unsatisfied due to limited accumulation in tumor site. Higher doses were administered to improve treatment efficacy, while severe side effects were caused. Therefore, we have revised this description to "their implementation has been restricted by low treatment efficacy, due largely to the absence of tumor-specific accumulation following administration" in the Abstract.

3. In the introduction, the authors described "which separately achieves 2- and 5-fold higher colonization in tumor tissue at 12 and 60 hours post tail vein injection in mice". However, the result in figure 4b showed higher tumor accumulation at 36 h and 60 h. Please double check it.

Response: We have used plate counting and IVIS to determine the colonization of bacteria in tumor tissue. The bacteria separately achieved 2- and 4-fold higher colonization in tumor tissue at 12 and 60 hours, which were quantified by bacterial counting (Figure 4c and d). The results in Figure 4b were measured by IVIS imaging, which was an indirect reflection of bacterial accumulation. These have been clarified in the caption of Figure 4.

4. Why the bacteria count is higher in EcN group than 5ApCB group in figure 4d.

Response: We apologize for this typo. The label has been switched in Figure 4d in the revised manuscript. Bacterial count of 5ApCB group was higher than that of EcN group.

5. Please provide the quantitative analysis of figure 6b.

Response: According to the reviewer's request, the quantitative analysis of Figure 6b was added in Figure S9 in the revised Supporting Information.

6. In the schematic diagram, the "cancer cell" should be "cancer cells".

Response: This has been addressed accordingly.

7. The "Figure 4b" in the manuscript is missing.

Response: This has been added in Line 21, Page 11 in the revised manuscript.

8. Please give out how the fluorescence intensity was quantified in figure 4b by in vivo data.

Response: We are grateful to the reviewer for drawing our attention to this issue. The details for quantification have been supplemented in the legend of Figure 4b.

Reviewer: 2

In this work, Geng et al reported the conjugation of aptamers to bacterial surface by a simple and cytocompatible amidation procedure, which could improve the localization of bacteria in tumor site following tail vein injection. The authors found that the conjugation of aptamer to the surface of bacteria increased its stability in serum. By varying feed ratio, the optimized bacteria conjugated with a certain number of aptamers per cell showed the highest specificity to tumor cells. After systemic injection, the conjugated bacteria generated a few times higher accumulation in tumor site in contrast to unmodified bacteria. The authors further implemented this approach with attenuated Salmonella, a strain that has entered phase I clinical trial. In both 4T1 and H22 tumor-bearing mice, aptamer-conjugated attenuated Salmonella presented significantly enhanced treatment efficacy, along with highly activated intratumoral immune responses. Overall, the conjugation of living bacteria with aptamers to improve their localization in tumor site is very innovative, and the conclusions are well supported by the in vitro and in vivo data. The findings would be of great interest to researchers in the field of drug delivery and cancer therapy. Therefore, the reviewer recommends the publication of this manuscript in Nature Communications provided the below minor issues have been appropriately addressed.

Response: We thank the reviewer very much for her/his positive review of our manuscript.

1) Compared to 4T1 cells, EcN and 5ApCB were separately added to the culture medium of 293T cells and the results showed no significant difference in fluorescence intensity between the binding of 293T cells with EcN and 5ApCB. The explanation of the mechanism was missing and should be added.

Response: The reason for no significant difference in fluorescence intensity between the binding of 293T cells with EcN and 5ApCB was the limited expression of Nucleolin on the surface of 293T cells (Figure S2b).

2) The authors mentioned in the manuscript that the mice injected with 5ApCB showed the strongest fluorescence signal, which could extend to 60 hours post injection. However, the authors did not describe the details on how to capture the fluorescence in tumor imaging section, as which is critical for the assessment of targeting effect.

Response: The details for quantification have been supplemented in the legend of Figure 4b.

3) In both 4T1 and H22 tumor-bearing mouse models, the bodyweight of mice dropped in the first 2 days post-injection. Was this caused by the injection of too much bacteria? How to decide the dose of bacteria? These should be clarified.

Response: Despite the removal of virulence factor, treatment with VNP20009 can cause bodyweight loss of mice, which has been reported commonly in the literature. The dose of bacteria used in our in vivo experiments was based on previous studies, which suggest that a single intravenous injection of VNP20009 at a dose ranging from 1×10^4 to 1×10^6 CFU/mouse is tolerable. This has been clarified in Line 15, Page 13.

4) In Figure 5g, it was claimed that histology images of the sampled tumor tissues after treatment with conjugated bacteria suggested the largest damaged area. The damage areas need to be pointed out in the H&E images and the description should be added to the figure legend.

Response: According to the reviewer's suggestions, the damaged areas have been circled by white dotted line and the details have been added to the legend (Figure 5i).

5) Figure 6b included the flow cytometric analysis data of H22 cells after co-incubation with PBS, VNP, and T-5ApCB at 37 °C for 1 hour. The geometric mean fluorescence intensity of H22 cells after co-incubation with PBS, VNP, and T-5ApCB should be added.

Response: The geometric mean fluorescence intensity of figure 6b was added as Figure S9 in the revised Supporting Information.

6) In the section of enhanced anticancer efficacy of ApCB: VNP were similarly conjugated with AS1411 and the antitumor efficacy of the resulting 5ApCB was assessed in vivo. Although 5ApCB showed significantly enhanced antitumor efficacy in comparison with VNP group, the binding efficiency of VNP and 5ApCB with 4T1 cells should be examined in vitro.

Response: Basing on the reviewer's request, the binding efficiency of VNP and 5ApCB with 4T1 cells was examined by in vitro assay and the results shown in Figure S5 and S6 indicated similar increment in binding after decoration with AS1411.

7) The legends of Figure 5 should be consistent with its contents. Images of tumor tissues stained with (g) TNF- α and (h) TUNEL. Scale bar: 50 μ m. (i) Typical H&E staining images of the sectioned tumors after different treatments. Scale bar: 200 μ m.

Response: These have been corrected in Figure 5 in the revised manuscript.

Reviewer: 3

In the research article, Aptamer-assisted tumor localization of bacteria for enhanced biotherapy,

the authors demonstrated the design, synthesis, and efficacy of an anticancer biotherapy based on non-virulent gram-negative bacterial strains that have been chemically modified by the conjugation with DNA aptamers to the cell surface. The authors showed that this system has increased efficacy against in vivo cancer models than equivalent, non-targeted, treatments. Additionally, the authors showed that the density of aptamers on the cell surface required optimization, as too many aptamers led to a reduced binding efficiency by the conjugated aptamers. While the work of some interest, the main issues that this article is facing is a lack of control experiments, a need to expand on their calculations in some areas, and a general need for the language to be cleaned up in some areas.

Response: We thank the reviewer very much for offering insightful comments on how to further refine the conclusions of our work.

1) Manuscript requires extensive review as grammatical errors can interfere with reader comprehension.

Response: The English has been polished throughout the manuscript.

2) Scrambled sequences of the same lengths as aptamers should be used in all experiments to assess the effect of nucleic acid modifications.

Response: We are grateful to the reviewer for highlighting this issue. Per her/his request, CTL, a scrambled oligonucleotide with the same number of bases but without secondary structure, was used as a control. As expected, the cellular binding efficiencies of CTL in these experiments showed insignificant differences compared with undecorated bacteria (Figure 2 and 4; Figure S2a, S4, and S5).

3) Please clarify the calculations/equations used to determine the number of aptamers/cell from spectrofluorimetry results and flow cytometry.

Response: According to the reviewer's suggestion, the calculations/equations used to determine the number of aptamers/cell has been added in the revised Supporting Information (Line 15, Page 4). First, several Cy5-labelled AS1411 solutions with certain concentrations were prepared. Then, a standard curve was established by plotting fluorescence intensity against Cy5-labelled AS1411 concentrations with the help of photoluminescence spectroscopy using excitation at 647 nm and emission at 670 nm. After determining the regression equation for the standard curve, the concentrations of free Cy5-labelled AS1411 could be calculated after reaction. The average number of aptamers per bacterial cell was calculated as following:

$$N_{avg} = ((C_0V - C_aV) \times NA) / N_g$$

where N_{avg} is the average number of aptamers per bacterial cell, C_0 is the concentration of free Cy5-labelled AS1411 before reaction, C_a is the concentration of free Cy5-labelled AS1411 after reaction, v is the volume that the reactions took place in, NA is the Avogadro constant 6.02×10^{23} , N_g is total number of bacterial cells.

4) It would be highly beneficial to the manuscript to provide individual names for each treatment so that they can be clearly distinguished from each other and the effects being described can be easily identified. For example, on page 11, when describing IVIS imaging, it is unclear what is fluorescing in the system as written.

Response: As suggested, the individual names have been provided for each treatment in the revised manuscript.

5) The data from the PBS treatments needs to be shown in all cases. The same procedures which were followed for the other two treatments (non-conjugated and aptamer-conjugated) need to be followed and the results shown for the PBS treatment. For example, the bacterial culture counting plates for bacteria isolated from tumor tissues post-treatment in mice should be shown. This is an issue in all experiments comprising these three groups (PBS, Bacteria, Aptamer-Bacteria).

Response: Per the reviewer's request, the data from the PBS treatments under the same procedures have been supplemented in the experiments of bacterial plate counting including bacteria isolated from tumor tissues post-treatment for the comparison between non-conjugated and aptamer-conjugated bacteria. The results were shown in Figure 2 and 4 as well as Figure S2a, S5, and S6, which further refined the conclusion that the increment was ascribed to the conjugation with aptamers.

6) It would be beneficial to repeat the studies demonstrating the optimal surface density of the aptamers for each bacterial strain tested, instead of just the EcN.

Response: According to the reviewer's suggestion, *Salmonella Typhimurium* VNP20009 (VNP) with different surface densities of aptamers were prepared (Figure S4). In vitro binding of VNP onto 4T1 cells was examined and the flow cytometric analysis showed that 5ApCB_{GFP} presented the highest affinity (Figure S5 and S6), which was consistent with the results of EcN strain.

7) Additional background information regarding the TLS11a aptamer (1-2 sentences) would be helpful.

Response: Additional background information regarding the TLS11a aptamer has been added in Line 10, Page 15 in the revised manuscript.

8) Is there evidence that the amine-Aptamer only interacts with the cell walls or will this system react and covalently bond to any exposed carboxylic acid groups (e.g. on glutamate or aspartate residues as well).

Response: As a hydrophilic macromolecule, it is difficult for aptamers to permeate cellular and nuclear membranes by passive diffusion. Therefore, the amide condensation may mainly happen on the surface of bacteria. At same time, a large number of free carboxyl groups are exposed on the outer-most surface of structured surface layers adhering to the rigid cell wall. Furthermore, carbodiimide activation of α - and β -carboxyl groups on the structured surface layers shows higher reactivity than those from glutamate or aspartate residues (Methods Enzymol. 1972, 25, 616; Prog. Biophys. Mol. Biol. 1988, 51, 131; Journal of Bacteriology 1989, 171, 5296). This has been clarified in Line 16, Page 6.

9) Figure 4. The color scheme flips between each image. Please keep this consistent between panels. Additionally, the caption does not match for figure 4. This is also true for Figure 5. Please add, if possible, the data from the PBS treated mice in this and all subsequent figures.

Response: According to the reviewer's suggestions, the color scheme and caption have been

updated in Figure 4 and 5. The PBS treated group has been added in Figure 2 and 4 as well as Figure S2a, S4, and S5.

10) Preparation of ApCB-Please clarify the volume that the reactions took place in and in the corresponding discussion section, please add a brief statement on the role of EDC and NHS in the reaction.

Response: The volume of the reactions took place in was 1 mL. In amide condensation, 1-ethyl-3-(3-dimethylaminopropyl) carbodiimide (EDC) was able to active carboxyl group forming an unstable O-acylisourea. Adding N-hydroxysuccinimide (NHS) to the reaction mixture could stabilize the activated carboxyl groups and subsequently improve the conversion of amide condensation. These have been stated in the revised Supporting Information (Line 8, Page 4).

11) Serum Stability-Please include more details regarding the experiment assessing serum stability. Cy5 should fluoresce regardless of the degradation status of the aptamer, so please clarify how the assay as described demonstrates the serum stability of the aptamer.

Response: All these samples were centrifuged and washed with PBS to remove any degraded aptamers before measurement. The details of experiments regarding serum stability assessment have been added in the revised Supporting Information (Line 7, Page 5).

12) PBS is used interchangeably to refer to phosphate-buffered saline as well as phosphate-buffered serum

Response: PBS refers to phosphate-buffered saline in this manuscript. This has been clarified in Line 20, Page 6.

13) The “severe side effects” of bacterial-mediated biotherapies are not mentioned but in Figure 1b, TNF and INF are illustrated. Please elaborate on these side effects and the significance of the illustrated cytokines.

Response: According to our previous studies, dosing with bacteria could induce acute inflammatory responses in mice (Nat. Commun. 2019, 10, 3452). In tumor tissue, tumor necrosis factor- α (TNF- α) is an important indicator of antitumor immune response and interferon- γ (IFN- γ) is crucial for intracellular immunity against tumor. Significant increments in the expressions of IFN- γ and TNF- α were observed in tumors from mice treated with 5ApCB, suggesting a reduction in T-cell exhaustion and an activation of T-cell antitumor activity. This has been clarified in the revised manuscript (Line 20, Page 14).

14) Pg. 10: Authors discuss the intravenous dosing of VNP, yet the corresponding confocal images are that of EcN

Response: Mice were intravenously injected with EcN. We are sorry for this typo, which has been corrected in the revised manuscript (Line 1, Page 10).

We thank all the reviewers again for taking their valuable time to review our manuscript. Their kind help and useful inputs are highly appreciated.

REVIEWERS' COMMENTS

Reviewer #1 (Remarks to the Author):

I really appreciated the authors significant efforts to address the various issues raised by the reviewers. Now the authors have generated a much stronger manuscript. I have one more question for the author consideration before publication of the current manuscript in Nature Communications. There is still a major concern for the safety issue for the bacteria mediated tumor therapy. The authors may consider a more comprehensive evaluation for a relatively longer period of time and discussion in a broader aspect, hopefully consideration of the therapeutic application of bacteria in the past.

Reviewer #2 (Remarks to the Author):

In this revised manuscript, the authors provided additional description and data to clarify the experiment procedures and conclusions. Overall, the reviewers' concerns have been addressed appropriately.

Reviewer #3 (Remarks to the Author):

All my comments were addressed

Reviewer: 1

I really appreciated the authors significant efforts to address the various issues raised by the reviewers. Now the authors have generated a much stronger manuscript. I have one more question for the author consideration before publication of the current manuscript in Nature Communications. There is still a major concern for the safety issue for the bacteria mediated tumor therapy. The authors may consider a more comprehensive evaluation for a relatively longer period of time and discussion in a broader aspect, hopefully consideration of the therapeutic application of bacteria in the past.

Response: Since the beginning of bacteria used for tumor treatment, the clinical safety was an important issue. In our pervious study, the bacterial number within healthy tissues decreased to a very low level at day 12 post-injection, which in turn remained at a significant high level in tumor tissue (Nat. Commun. 2019, 10, 3452). In this work, the levels of inflammatory reactions caused by bacteria were similar to those of PBS group at 60 h post-injection. These results demonstrated the safety of bacteria mediated tumor therapy. Actually, bacteria mediated treatments have attracted significant attention for more than one hundred years. Especially in recent decades, bacteria mediated tumor therapy had been applied in several clinical trials. For instance, *S. typhimurium* VNP20009 has been tested in several phase I trials. A phase I trial displayed that all patients with metastatic melanoma were safely administered the maximum tolerated dose of VNP20009 intravenously in 30 minutes (J. Clin. Oncol. 2002, 20, 142-152). To increase the delivery amount of VNP20009 to solid tumor, four patients were given bacterial infusion for 4 hours, demonstrating the safety of bacteria mediated tumor therapy (J. Immunother. 2003, 26, 179-180). In another phase I clinical trial, three patients received bacterial intratumoral injection of *S. typhimurium* VNP20009 with *E. coli* CD gene for cancer therapy. The engineered bacteria displayed acceptable safety in all patients and showed enhanced antitumor efficacy in two of them (Cancer Gene Ther. 2003, 10, 737-744). Currently, *L. monocytogenes* have displayed promising clinical results in patients. Up to now, three phase II trials of engineered *Listeria* strains have been accomplished (J. Clin. Oncol. 2015, 33, 1325-1333; Gynecol. Oncol. 2020, 158, 562-569; <https://clinicaltrials.gov/ct2/show/NCT02853604>).

Reviewer: 2

In this revised manuscript, the authors provided additional description and data to clarify the experiment procedures and conclusions. Overall, the reviewers' concerns have been addressed appropriately.

Response: We thanks the reviewer for her/his agreement on our revised manuscript.

Reviewer: 3

All my comments were addressed

Response: We thank the reviewer for her/his positive feedback.